# Impact of variability in cell cycle periodicity on cell population dynamics

**Chance M. Nowak[1,2,3], Tyler Quarton[1,2], Leonidas Bleris[1,2,3]***

**1** Bioengineering Department, The University of Texas at Dallas, Richardson, Texas, United States of America, **2** Center for Systems Biology, The University of Texas at Dallas, Richardson, Texas, United States of America, **3** Department of Biological Sciences, The University of Texas at Dallas, Richardson, Texas, United States of America

\* bleris@utdallas.edu

**Data Availability Statement:** All data and the computer code used to perform the analyses in this article is available in the following GitHub repository: https://github.com/BlerisLab/

## Abstract

The cell cycle consists of a series of orchestrated events controlled by molecular sensing and feedback networks that ultimately drive the duplication of total DNA and the subsequent division of a single parent cell into two daughter cells. The ability to block the cell cycle and synchronize cells within the same phase has helped understand factors that control cell cycle progression and the properties of each individual phase. Intriguingly, when cells are released from a synchronized state, they do not maintain synchronized cell division and rapidly become asynchronous. The rate and factors that control cellular desynchronization remain largely unknown. In this study, using a combination of experiments and simulations, we investigate the desynchronization properties in cervical cancer cells (HeLa) starting from the $G_1$/S boundary following double-thymidine block. Propidium iodide (PI) DNA staining was used to perform flow cytometry cell cycle analysis at regular 8 hour intervals, and a custom auto-similarity function to assess the desynchronization and quantify the convergence to an asynchronous state. In parallel, we developed a single-cell phenomenological model the returns the DNA amount across the cell cycle stages and fitted the parameters using experimental data. Simulations of population of cells reveal that the cell cycle desynchronization rate is primarily sensitive to the variability of cell cycle duration within a population. To validate the model prediction, we introduced lipopolysaccharide (LPS) to increase cell cycle noise. Indeed, we observed an increase in cell cycle variability under LPS stimulation in HeLa cells, accompanied with an enhanced rate of cell cycle desynchronization. Our results show that the desynchronization rate of artificially synchronized in-phase cell populations can be used a proxy of the degree of variance in cell cycle periodicity, an underexplored axis in cell cycle research.

## Author summary

The cell cycle is the series of events that a cell undergoes to replicate its DNA and divide into two identical daughter cells. Blocking and synchronizing cells in the same phase is an invaluable tool for studying the properties and associated biology of the cell cycle.

ProjectCellCycle2023. All computer code was written in the MATHEMATICA programming environment.

**Funding:** LB acknowledges funding from the US National Science Foundation (NSF) grants (1351354, 2029121, 2114192), a Cecil H. and Ida Green Endowment, and the University of Texas at Dallas. The funders had no role in study design, data collection and analysis, decision to publish, or preparation of the manuscript.

**Competing interests:** The authors have declared that no competing interests exist.

Intriguingly, when synchronized cells are released, they rapidly become asynchronous, but the factors that control this process remain largely unknown. In this study, we investigated how cells become desynchronized after being synchronized using a common laboratory technique used to halt cell cycle progression. We developed a single-cell mathematical model that returns the DNA amount across the cell cycle stages and fitted parameters using experimental data. Simulations of cell populations revealed that the rate of cell cycle desynchronization is primarily determined by the variability in the length of the cell cycle within a population, which result was subsequently validated experimentally. Our study demonstrates that the rate of desynchronization can be used as a proxy for the degree of variance in cell cycle periodicity, which is an underexplored axis in cell cycle research.

## Introduction

Cell division is traditionally described as a general process divided into two phases, the interphase and mitosis (cell division). Interphase is further divided into three subphases; Gap 1 phase ($G_1$) in which the cell has a DNA content of 2n, synthesis phase (S) in which the cell's DNA content is greater than 2n but less than 4n, and Gap 2 phase ($G_2$) in which the cell's DNA content is 4n upon completion of synthesis. Early observations into cell cycle progression showed that the timing of $G_1$ phase is highly variable not just between cell types but also between cells within a monoclonal population, and that this variable length directly impacts the heterogeneity observed in clonal populations for cell cycle periodicity [1,2]. Additionally, a critical point in the cell cycle was discovered [3], in which cells were found to be committed to DNA synthesis independent of environmental factors. Moreover, it was later demonstrated that under various suboptimal nutritional conditions, cell cycle progression could be arrested at the $G_1$/S boundary, and escapement into S-phase could only occur once suitable nutritional needs were restored [4]. The boundary was termed the restriction point (R-point), whereby cells could enter a lower metabolic rate (a quiescent state) to remain viable until adequate nutrition is restored allowing the necessary constituents to be present in suitable amount to enable DNA synthesis [4]. Ultimately, it was shown that the high variability of $G_1$ phase duration can be attributed to a cell's ability to overcome the restriction point [5].

Investigations into cell cycle progression and regulation often start with the need to synchronize cells within a population to the same cell cycle phase [6,7]. One common approach to cell cycle synchronization is the double-thymidine block that interferes with nucleotide metabolism resulting in an inability of the cells to synthesize DNA causing a cell cycle arrest at the $G_1$/S boundary [8,9]. Interestingly, when synchronized cell populations are released from cell cycle arrest, they quickly desynchronize, and reach a state of "asynchronicity," whereby the individual cell cycle phases stabilize into fixed percentages within the overall population. Indeed, simply sampling cells from an asynchronously growing *in vitro* cell culture will reveal (Fig 1a) the fixed percentages for the three phases of interphase ($G_1$, S, and $G_2$). Additionally, cells can be pulse-labeled with bromodeoxyuridine (BrdU) to create a semi-synchronous cell population in which only cells in actively progressing through S-phase incorporate the thymidine analog BrdU into their genome, and thus the original pulse-labeled population can be tracked overtime by using a fluorescently conjugated BrdU antibody [10]. These observations again showed that the initially pulse-labeled cells progressed synchronously through the cell cycle for some time before

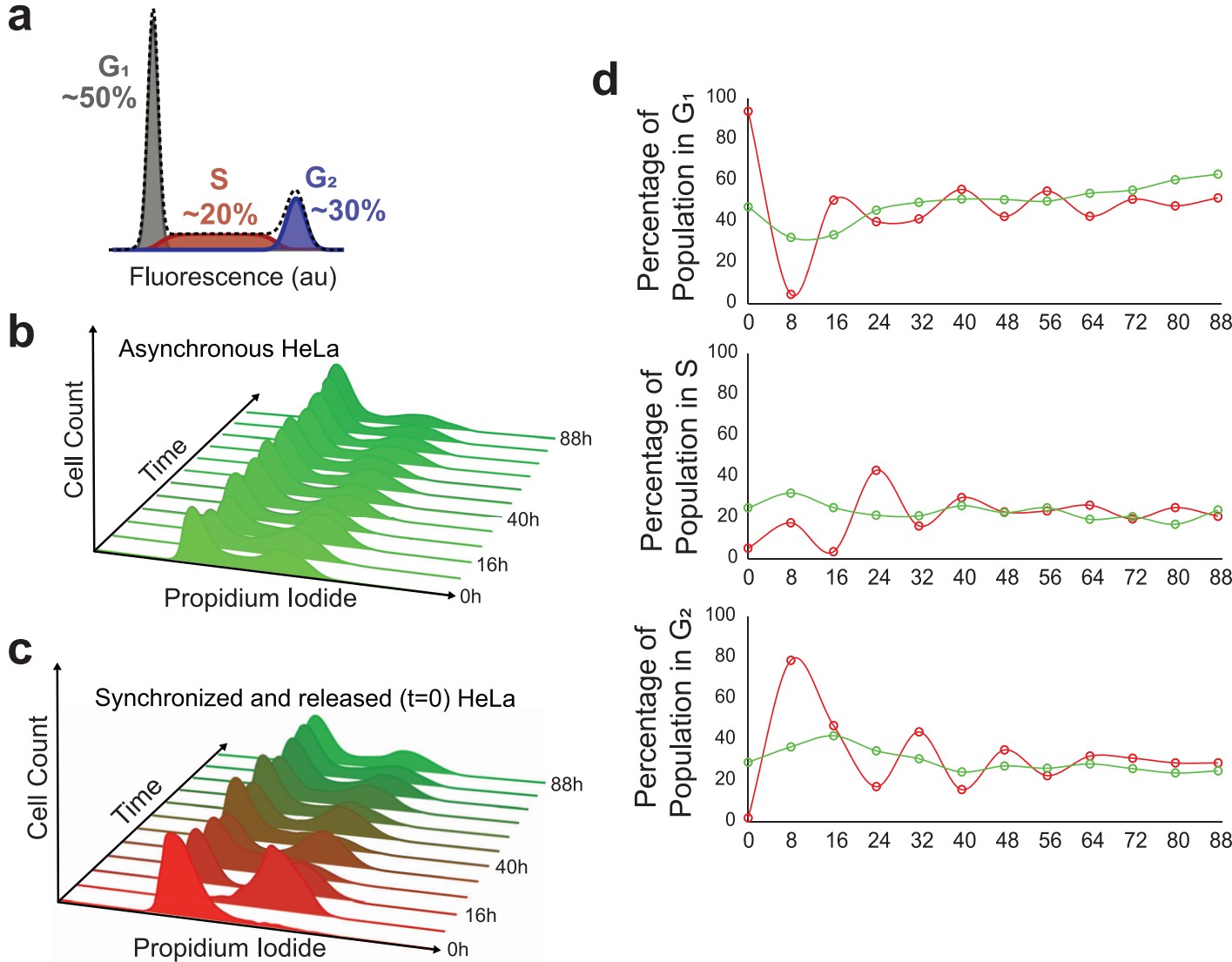

**Fig 1. Cell desynchronization via double thymidine block and release. a)** Cell cycle phases as indicated by cell DNA content and approximate phase distribution in an asynchronous population. **b)** Fluorescent profile of propidium iodide (PI) stained cells during asynchronous growth from t = 0 to t = 88. **c)** Fluorescent profile of PI-stained cells following $G_1/S$ synchronization by double thymidine block from t = 0 to t = 88. **d)** Percentages of cells in a given cell cycle phase at a given time point; asynchronous cell growth in green and desynchronous cell growth in red. The cell cycle phase percentages for each time point were determined via the Dean-Jett-Fox model.

quickly desynchronizing and resorting back to an asynchronous DNA distribution profile.

The inherent variability of cell cycle duration between identical cells may be accounted for by considering sources of cellular noise. In other words, the variability between cellular constituents such as signaling and transcriptional factors, along with the biochemical stochasticity of molecular interactions do likely propagate to the phenotypic level and may be responsible for varying timing events that dictate cell cycle progression. For example, signalling factors in a tumor microenvironment that confer a higher degree of intercell variability contribute to tumor cell heterogeneity and pathology [11,12]. Therefore, it is important to examine the implications of cellular noise to cell cycle periodicity.

In this report, we investigated the rate of cell cycle desynchronization by measuring the change in the DNA distribution of a population of cells over time. To this end, we measured the single-cell DNA amount of a population of cells as they transition from an initial state of cell cycle synchrony, where cells are experimentally locked into the $G_1$/S boundary, to a state of asynchrony. We used statistical tools to quantify the dynamic change in the DNA probability density function over time from an initial synchronized cell population. Subsequently, we developed a mathematical model to simulate at single-cell level the DNA amount as the cell transitions through cell cycle states, and finally, experimentally validated our model prediction. More specifically, our model revealed that cell cycle desynchronization rates were particularly sensitive to the variability of cell cycle duration within a population. With this insight, to validate the results we introduced external noise in synchronized cells using lipopolysaccharide and, indeed, confirmed an increase in cell cycle desynchronization. Considering the ubiquitous role of the cell cycle properties to cell health, the implications of our work extend to numerous fronts further elaborated in the discussion.

## Results

### Thymidine-based arrest and desynchronization

The exogenous introduction of excessive thymidine into cells interrupts DNA synthesis, arresting the population of cells in the $G_1$/S-phase transition. Upon release, the population of cells are permitted to reenter their respective cell cycles. Ultimately, the population of cells will become asynchronous with respect to their cell cycles, yielding a PI fluorescent profile. The PI distributions dynamically change as the population desynchronizes.

After cells were synchronized via double-thymidine block, timepoints were collected every 8 hours for a total of 88 hours. Both asynchronous (untreated) cells (Fig 1b) and synchronized (Fig 1c) were subjected propidium iodide staining and flow cytometry analysis. Notably, we observed near full synchronization of cells as judged by the first few timepoints (Fig 1c) in the synchronous population. While inhibition of DNA synthesis can cause replicative errors due to stalled replication forks, resulting in quiescence or cell death, we did not observe either an increase in cell death nor any quiescent populations, which would manifest as a sub-$G_1$/$G_1$ population at timepoint 8. Each PI histogram was subjected to cell cycle phase classifier [13–15] with the cell cycle phase distribution displayed as percentages of the total population. As we observe in Fig 1d, the synchronized population eventually reaches an asynchronous distribution. The residual plots of the DNA distribution of the synchronous population against the asynchronous population ultimately converges to within 8.4%, 1.5%, and 6.1% of $G_1$, S, and $G_2$, respectively (S1 Fig).

### Quantifying cell synchronicity

The DNA dynamics during interphase of a population of cells are defined by the population's collective distribution of its DNA at a given time. If all the cells within a population are undergoing interphase synchronously, time separated measurements of the population's DNA distribution will accordingly change in time. This would mean that the DNA distribution of a population of cells will be different for each time measurement. Conversely, if the population's cells are independently progressing through interphase, temporal differences between the population's DNA distribution become indistinguishable, rendering its DNA distribution into a seemingly unchanging profile (Fig 1a).

With this in mind, we can create a set of assumptions: Let $\{\mathbf{X}_t\}$ denote sets of observations generated from an evolving probability distribution at any point in time $t$. We define the auto-

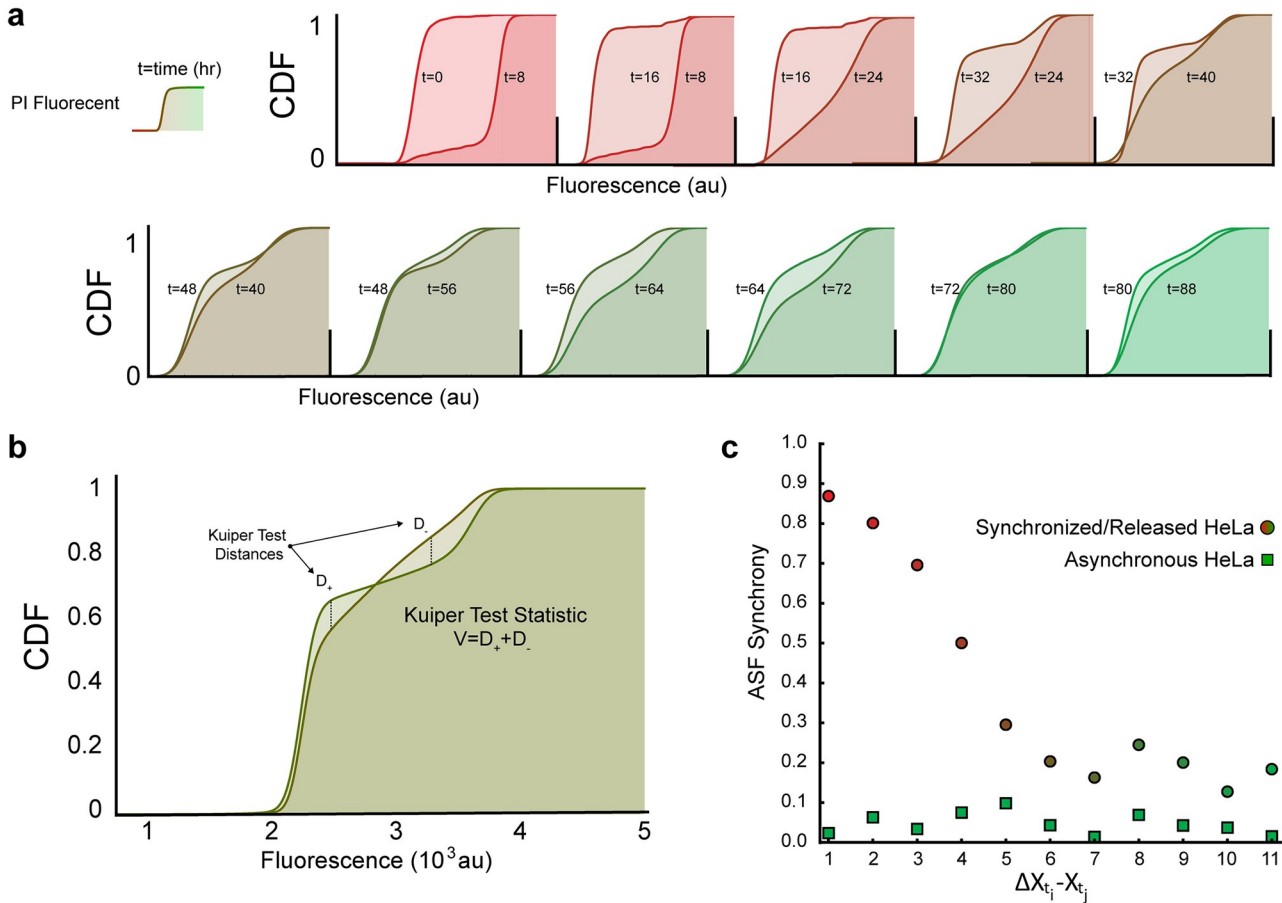

**Fig 2. Rate of desynchronization using Kuiper test statistic. a)** Pairwise comparison of PI CDFs for each time point (data shown is from synchronized cells). **b)** Visual representation of Kuiper Test Statistic determination between time points. **c)** Rate of desynchronization between asynchronous (green) and synchronized (red) Hela cells. Over time (~60 hours) synchronized cells being to reach an asynchronous state.

similarity function (ASF) between times $t_1$ and $t_2$ as

$$\boldsymbol{\Sigma}_{XX}(t_1, t_2) = \underset{-\infty < x < \infty}{max} \left( F_{X_{t_1}}(x) - F_{X_{t_2}}(x) \right) + \underset{-\infty < x < \infty}{max} \left( F_{X_{t_2}}(x) - F_{X_{t_1}}(x) \right) \quad (1)$$

where $F_{X_t}$ denotes the cumulative distribution function of a given set of observations $X_t$.

Essentially, the auto-similarity function is the Kuiper two-sample test statistic, which measures the similarity between two sets of data, performed on a single, time evolving variable $X_t$ rather than two distinct variables (Fig 2a). The Kuiper test statistic is rotation-invariant, making its application insensitive to the "starting points" of the data to be compared. As the DNA content measured in our cell populations cycle between 2n to 4n, the data collected from our cell cycle experiments are inherently cyclical, making the use of a rotation-invariance test statistic ideal (Fig 2b). If the evolving distribution eventually converges to a steady-state, we expect $\Sigma_{XX}(t_i, t_{i+1}) \to 0$ for some successive time measurements $t_i$ and $t_{i+1}$ as $t \to \infty$, where a value of 0 indicates full asynchrony. Conversely, we interpret non-zero, positive evaluations of the ASF to indicate dissimilarity, where, in the case of a cyclically evolving sets of data,

evidence that the underlying probability distribution is in a transient state, where a maximum value of 1 indicates full synchrony (Fig 2c).

In our experiments, $\{\mathbf{X}_t\}$ is variable DNA fluorescently measured by flow cytometry in PI-stained populations of cells, where $t_i = \{0,8,16,\ldots,88\}$ indicates the hour corresponding to the $i_{th}$ measurement of data collected with respect to their release from cycle arrest via double thymidine block at $t_0 = 0$. We expect that the ASF evaluation of times $t_0$ and $t_1$ to be the greatest as the population of cells synchronously progress through the cell cycle, resulting in markedly dissimilar distributions of DNA in observation sets $\mathbf{X}_{t_0}$ and $\mathbf{X}_{t_1}$.

As the individual cells within a population variably progress through the cell cycle, we expect population DNA distributions to diverge, eventually settling to the classic asynchronous distribution profile (Fig 1), where successive measurements of a no-longer-evolving variable are expected to be near-zero. We calculated the ASF between each temporally successive pair of data for both the synchronized cell population and the asynchronous control population and found that the ASF converges to a minimum of 0.127 from an initial value of 0.869, following a logistics curve. We observed an expected linear ASF from the asynchronous population with slight oscillations, most likely emerging from unintended loss of mitotic cells during harvesting (mitotic shake off) positive slope (Fig 2c).

## A single cell interphase model

Cell cycle progression is intimately linked to a cell's dynamically changing DNA content. Temporal transitions from a cell's state of 2n to 4n define cell cycle phases, where $G_1$, S, and $G_2$, correspond to genetic quantities of 2n, 2n+, and 4n, respectively, where the event of mitosis restarts the cell cycle for two progeny cells. Deterministically, we model a single cell's dynamic DNA content as

$$dna(t) = dna_0 + \frac{dna_{max} - dna_0}{1 + e^{-\beta(t-s)}} \quad t_0 \leq t \leq \tau \tag{2}$$

where $dna_0$ is the initial genetic content in phase $G_1$, $dna_{max}$ is the maximum genetic content after synthesis, $\beta$ parameterizes the synthesis rate, $s$ is the time in which the cell is halfway through synthesis and determines the periods of $G_1$, S, and $G_2$, and $t$ is time. Thus, the $\beta$ and $s$ variables account for the S phase of the cell cycle. We assume that synthesis faithfully duplicates the genetic content, where $dna_{max} = 2\,dna_0$, thus reducing Eq 2:

$$dna(t) = dna_0 \left(1 + \frac{1}{1 + e^{-\beta(t-s)}}\right) \quad t_0 \leq t \leq \tau \tag{3}$$

We can further reduce Eq 3 by representing $\beta$ and $s$ as functions of the cycle period $\tau$ as:

$$dna(t) = dna_0 \left(1 + \frac{1}{1 + e^{-24\left(t - \frac{2\tau}{3}\right)}}\right) \quad t_0 \leq t \leq \tau \tag{4}$$

where we initially assumed a period of 24 hours and the duration of S phase to be approximately 1/3 of the period, which is 8 hours, agreeing with previous reported values [16]. Accordingly, this single cell model captures DNA amount during interphase using two parameters, the initial DNA amount and the cell cycle period (Fig 3a).

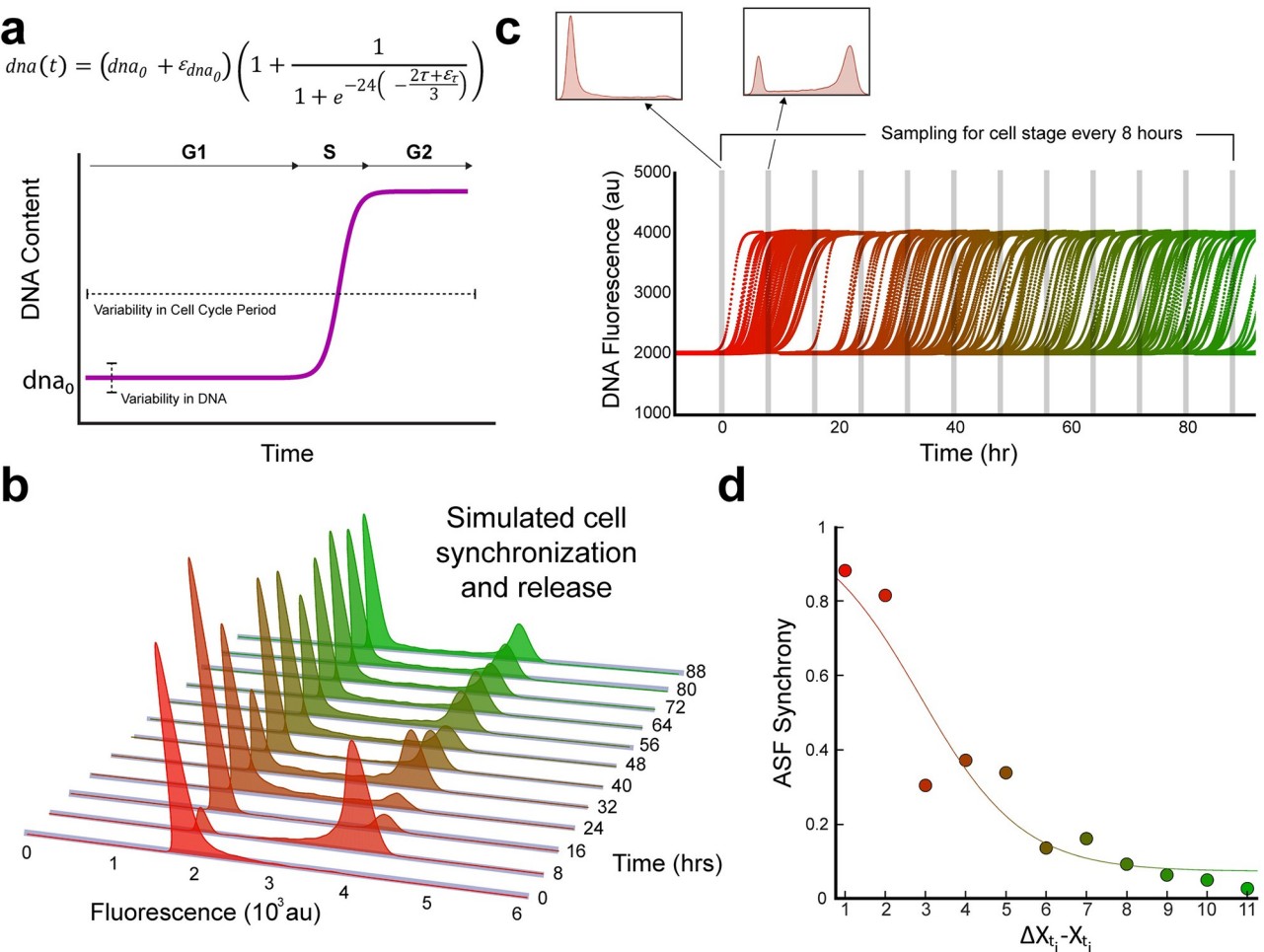

**Fig 3. Single cell model of desynchronization. a)** DNA synthesis is captured by the Gaussian error function where the relative durations of cycle phase are tunable. **b)** Simulated data of PI staining of multiple lineages with normally distributed initial gene content. **c)** Cell cycle pace inheritance following a Gaussian distribution. **d)** Desynchronization rate of simulated cell population.

To study the impact of the cell cycle period to the rate of asynchrony we use an Error-in-Variables (EIV) modeling approach [17,18] to add noise to the cycle periodicity:

$$dna(t) = \left(dna_0 + \varepsilon_{dna_0}\right)\left(1 + \frac{1}{1 + e^{-24\left(t - \frac{2\tau + \varepsilon_\tau}{3}\right)}}\right) \quad t_0 \leq t \leq \tau \tag{5}$$

where $\varepsilon_\tau \sim N\left(0, \sigma_\tau^2\right)$ is a normally distributed error term with variance $\sigma_\tau^2$ and $\varepsilon_{dna_0} \sim N\left(0, \sigma_{dna_0}^2\right)$ is additionally added to capture fluorescent variability seen as broadened peaks around $G_1$ and $G_2$. Thus, the variance is being applied to the initial DNA content, and the entire period (i.e., not applied to any given cell cycle stage). We simulate a population of 1,000 cells, each starting synchronously at $G_1$ with extrinsically varying initial DNA content and cell cycle periodicity, as they repetitively progress through interphase (Fig 3b). We then take temporal slices of the DNA content of the population of cells and plot the populations distribution of DNA content intermittently (Fig 3c). We finally apply ASF to the slices in a pairwise manner as performed with the experimental data (Fig 3d). We also explored the ASF

output of our model by comparing the EIV modeling approach with increasing periodicity noise between Poisson and normal distribution of both the DNA content and periodicity (S2 Fig). We observed that the Poisson distributed error term for DNA content and/or period, as opposed to a normal distribution, failed to reproduce ASF trends from synchronized cells. We also compared the effects of differing means on desynchronization rates, and our model revealed that there was no significant impact on desynchronization rates between 22-, 24-, and 26-hour periods each with a variance of 3 hours (S3 Fig). Importantly, we found that only by including a variance term to cell cycle periodicity were we able to capture population dynamics that recapitulate the experimental results. Moreover, our model revealed that increasing the magnitude of variance resulted in increasing rates of desynchronization (S4 Fig). Next, in order to further evaluate our model's prediction, we sought to assess cell cycle desynchronization by introducing an exogenous means to perturb the cell cycle dynamics.

## Impact of LPS on cell cycle duration variability

Lipopolysaccharide (LPS) is a major component of the outer membrane of Gram-negative bacteria that can bind to TLR4 receptors initiating a signaling cascade that ultimately results in NFkappaB translocation from the cytoplasm to the nucleus, where as a transcription factor, it initiates the upregulation of inflammation regulatory genes [19–21]. Additionally, NFkappaB activation can be induced by cytokines such as TNFalpha [22], which has been reported with contrasting roles, whereby NFkappaB induction is associated with both the activation of pro-survival genes as well pro-apoptotic genes [23]. In addition to regulating inflammation signaling pathways, NFkappaB regulates major cell cycle regulatory factors [24–27]. Interestingly, components of NFkappaB, such as RelA, have been shown to interact with key cell cycle regulators, such as E2F transcription factors that are crucial in controlling progression through the $G_1$/S boundary [25].

We therefore hypothesized that the contrasting nature of LPS stimulation in HeLa cells would result in a greater variance in overall cell cycle duration. Accordingly, if LPS is a viable approach for introducing cellular noise we would expect the desynchronization rate to increase compared to untreated synchronized cells (S5 Fig). Thus, in order to determine if LPS simulation had any effect on cell cycle duration, we conducted a time-lapse experiment to track individual cells cell cycle duration (Fig 4a). In order to have a better indication of relative position of each cell in relation to the cell cycle, we integrated a fluorescence tracker using lentiviral transduction that express the histone protein H2B fused to a fluorescent protein (H2B-FT) [28]. Upon expression, the H2B protein is incorporated into nucleosomes, which binds DNA, and therefore could more easily distinguish cells undergoing mitosis. Doubling time, cell growth and viability was assessed via Trypan Blue staining to assess any possible adverse effects on cell proliferation from lentiviral integration, of which none were observed (S6 Fig). Next, we treated asynchronously-growing HeLa cells with 1.0 μg/mL of LPS derived from *E. coli*. O111: B4, and monitored the duration of the cell cycle for individual cells with timelapse microscopy for 72 hours every 20 minutes (S7 Fig and S1–S4 Videos and S1 Data). We found that the overall variance was higher in treated cells versus untreated cells with an accompanying increase in the mean duration 23.7±4.73 and 21.7±3.42 hours, respectively (Fig 4b).

We next sought to test if the predictability of our simulation model with the values obtained from the time-lapse microscopy would result in an increased desynchronization rate under LPS stimulation. In order to compare multiple synchronous cell samples, we normalized each sample to its initial ASF value ($\mathbf{X}_{t_0} - \mathbf{X}_{t_8}$, S8 Fig). Upon inputting our new values obtained from the time-lapse microscopy, our model indeed predicted an increase in desynchronization when treated with LPS compared to the untreated sample (Fig 4c).

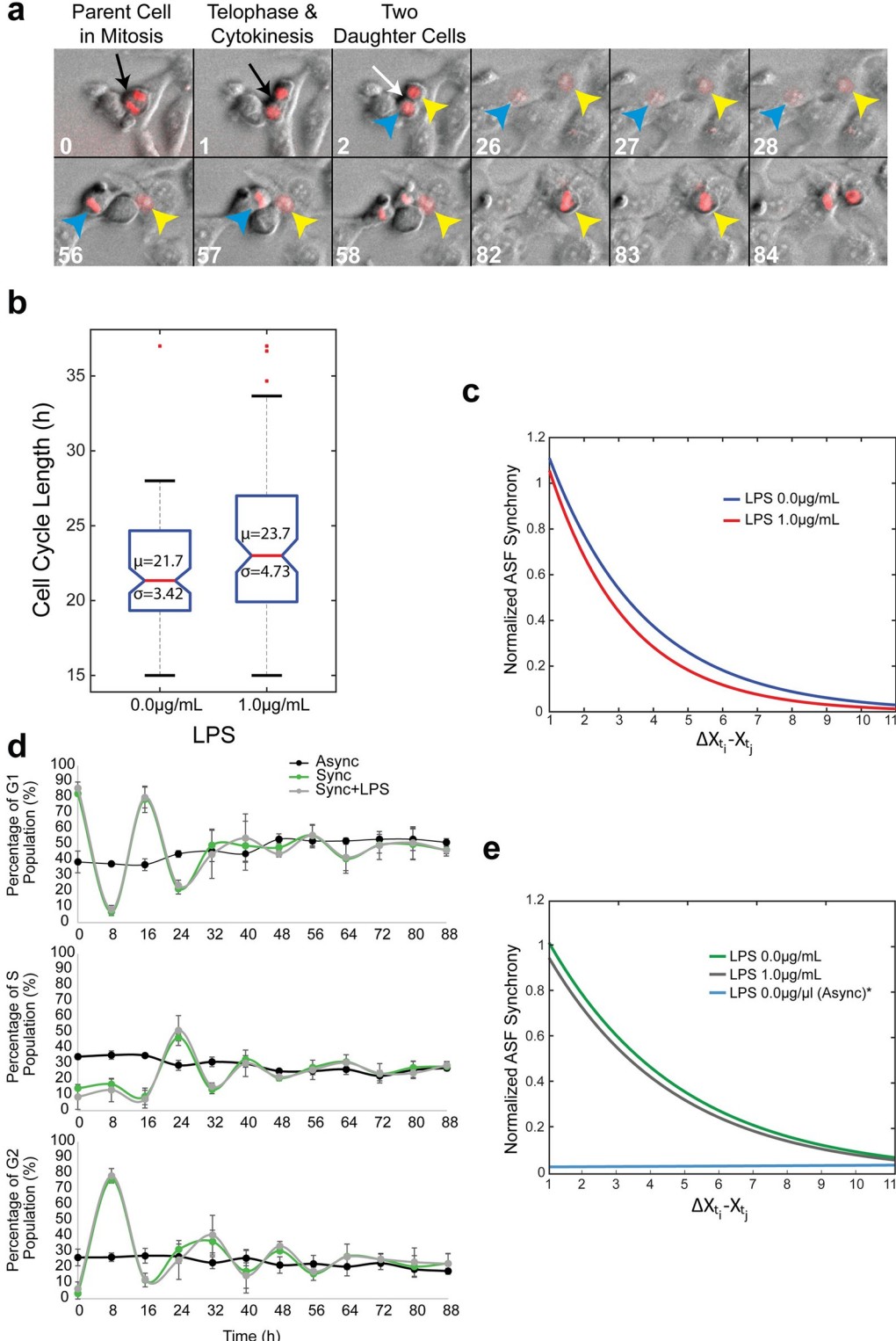

**Fig 4. Noise variation of cell periodicity. a)** Representative images of time lapse experiments. 100 cells were tracked for each condition and the population mean and standard deviation of cell cycle duration was determined. Once the septum (white arrow) is visible following cytokinesis, the cell cycle duration recording begins for both daughter cells (yellow and blue arrow). Both cells being cell cycle at Frame 2, and both daughter cells can be seen progressing through interphase in Frames 26–28. By the end of Frame 57, the first daughter cell completes the cell cycle and recording ends. The second

daughter cell (yellow arrow) had a substantially longer cell cycle duration, which concluded at the end of Frame 84, thus demonstrating the inherent variability of cell cycle duration between identical cells within the population. **b)** Asynchronous cells were wither treated with 1 μg/mL of LPS or left untreated and cell cycle duration was recorded (n = 100). **c)** Values obtained from time lapse microscopy for cell cycle mean and standard deviation were used in our model to predict the impact on cell cycle desynchronization. The model revealed the LPS administration should result in an increased rate of cell cycle desynchronization **d)** Cell cycle phase distribution of LPS treated cells following cell cycle synchronization for 88 hours post release (n = 3). **e)** Normalized ASF scores for LPS-treated desynchronizing cells. The asynchronous population was not normalized in order to capture the overall linear trend (n = 3).

Given that we were able to increase the variance of cell cycle duration with LPS, and that our simulation model predicted an increase in desynchronization due to increased cell cycle duration variance, we next tested if we could experimentally obtain higher rates of desynchronization using the previous approach of PI-staining time-separated synchronized cells. Therefore, we again synchronized HeLa cells via the double-thymidine block method, and immediately following release of the arrested cells, we treated with LPS 1.0 μg/mL and collected timepoints every 8 hours for 88 hours. We then analyzed the PI-stained cell populations via FlowJo cell cycle classifier that uses the Dean-Jett-Fox algorithm to observe the expected cell cycle state dampening oscillations towards asyncronicity (Fig 4d). Interestingly, our ASF analysis methodology was able to reveal the impact on cell cycle desynchronization with an enhanced rate of desynchronization from LPS treated cells (Fig 4e and S9 Fig).

## Discussion

There is a multitude of approaches to mathematical modeling of cell cycle dynamics and cell behavior. A differential equations approach will typically model the change in concentrations of the various molecules in cell cycle function over time [29,30]. There are also Boolean network models that represent the cell cycle as nodes and connected edges that are on/off switches and can be changed by specific molecular interactions [31,32]. Additionally, there are agent-based models that simulate individual cells to reveal information about populational cell behavior [33]. Stochastic models that capture the random fluctuations in the cell cycle that can be attributed to molecular noise, have been used to reveal how variability in individual cells impacts the dynamics of a population of cells [32,34]. Herein, we developed a single-cell phenomenological model the returns the DNA amount across the cell cycle stages and fitted the parameters using experimental data.

The cell cycle and subsequent daughter cell division is a central facet of cell biology from development and cellular differentiation to disease initiation and progression. Cell synchronicity is an essential aspect of mammalian biological homeostasis. The circadian rhythm is a molecular orchestrated process present in various tissues that synchronizes biological outputs to the 24-hour day-night cycle [35]. It is composed of multiple master transcription factor regulators that are involved in robust feedback networks [36].

While the cell cycle is tightly regulated and robust in a single cell, across a population we observe significant variability in period. Each cell within a given population contains measurable variations in their cellular content and housekeeping genes (e.g., differences in their RNA polymerases, ribosomes). These variations impact the expression of genes in what is known as extrinsic noise [17,37–42]. Furthermore, the cellular machinery responsible for progressing a cell through its cycle is intrinsically stochastic. Cellular noise occurs in genetically identical cells that exhibit variations in biochemical activity, and this inherent heterogeneity can manifest into observable phenotypes within a population of cells [43]. Indeed, intra and inter-cellular differences cause an initially synchronously in-phase population of cells to diverge as each progresses independently through their life cycle at varying rates [44].

Our approach offers a novel method that could potentially be utilized for ascertaining the overall noise of an engineered cell line compared to the parent cell line. Moreover, it is crucial to not only develop novel methods for measuring noise, but to discover new small molecules that can impact noise to lead to more desired outcomes. For example, Dar et al. performed a screen for bioactive molecules that enhanced the gene expression noise of latent HIV, which reactivated the HIV and in-turn makes the virus more susceptible to antiviral drugs [45]. Indeed, our own results stress the importance of using small molecules to understand and perturb cellular noise.

Here, using a combination of simulations and experiments we show that the variability in cell cycle period directly impacts the rate of desynchronization in a population of cells. The next line of investigation will include studying the factors that contribute to this variability at a single cell level, and the distribution between intrinsic and extrinsic sources of noise. It is well known that tightly regulated processes can rely on stochastic variations [46–48] but it is also pertinent to study how disease states depend on noise and if noise itself can drive disease progression. For example, an intriguing hypothesis is that cancer cells [49] obtain benefit by having higher noise in cell cycle periodicity, which yields ultra-slow and fast diving cells. This could then lead to cell populations that are able to escape the effects drugs that target rapidly dividing cells or lead to highly proliferative cells that result in aggressive tumor formation, which can be more difficult to treat. Moreover, daughter cells that rapidly lose synchrony may experience external cell-stage-specific stressors at different points in their respective cell cycles shifting the population to a subset of cells that may support survival, such as insensitivity to DNA damaging agents due to shortened S phase duration. Interestingly, Gram negative bacteria that produce LPS have been shown to exacerbate inflammation in cervical cancer cells, as well as promote proliferation and invasion [50–53]. We believe that the implications of cellular noise in cell synchrony and cell periodicity opens an exciting path towards exploiting the variability in cell cycle period for therapeutic purposes.

## Methods and procedures

### Cell culturing and synchronization

HeLa cells were grown in Gibco DMEM supplemented with 10%FBS, 1X PenStrep, 2mM glutamine, and 1X Gibco NEAA and grown at 37˚C with 5% $CO_2$. 50,000 cells were seeded per well in 6 well plates. 24 hours post-seeding cells were treated with 2mM of thymidine for 19 hours after which the cells were washed with 1X PBS and given fresh complete media to release from the first thymidine block. The cells then incubated for 9 hours before receiving a second dose of 2mM of thymidine for 15 hours. Cells were washed with 1X PBS to remove thymidine before given fresh media to continue to grow unimpeded. Cells harvested at t = 0 were collected immediately following the second PBS wash. Additional wells were harvested every 8 hours for 88 hours. Asynchronous cells were harvested at same time as synchronized cells for each time point. Cells were harvested by washing with PBS, detached from the well with trypsin-EDTA (0.25%) for 3 min at 37˚C then quenched with fresh complete media. Harvested cells were pelleted at 1000RPM for 5 min at room temperature. The supernatant was removed and the cell pellet was resuspended in 1X PBS, then pelleted again at 1000RPM for 5 min at room temperature. The supernatant was removed and the cell pellet was resuspended in 1 mL of 70% ethanol and stored at 4˚C for a minimum of 24 hours to fix the cells.

LPS derived from *E. coli* 0E111 was reconstituted in PBS without Mg or Ca at a concentration of 1mg/mL. LPS solution was added directly to the cell culture media after replacing with fresh media initiating the release from the double thymidine arrested state.

## Propidium iodide staining

After fixation, cells were pelleted by centrifuged at 1000 RPM for 5 minutes at room temperature. The fixing solution was aspirated off the cell pellet, and resuspended in 1X PBS. Cells were counted for each sample, and then normalized to the lowest cell count for uniform propidium iodide (PI) staining across samples. The PI staining procedure was done according to manufacturer's directions (Propidium Iodide Flow Cytometry Kit, cat# ab139418).

## Cell cycle phase analysis

Stained cells were subjected flow cytometry using a BD LSRFortessa™ flow cytometer. PI fluorescence was excited with a 561nm laser and emission was detected using a 610/20 nm bandpass filter. Assignment of cell cycle phases were performed using the univariate modeling via the Dean-Jett-Fox algorithm with FlowJo 10.7.1.

## Lentiviral HeLa transduction for H2B-FT expression

The fluorescent tracker sequence was obtained from addgene (#157671) and cloned using primers P1: gaagagttcttgcagctcggtgac and P2: cagtagggtaccccggaattagatcgatctctcgacatcc. The amplicon was digested with restriction enzymes BsiWI and KpnI and inserted into the Lenti-CRISPRv2 (addgene #52961) backbone. The resulting plasmid was transfected into HEK293T cells along with pMD-VSVG and psPAX2 plasmids to generate viral particles that are released into the media. The media was aspirated two days post-transfection, and replenished with 5 mL of fresh media every day for three days. The 15 mL of harvested viral-containing media was passed through a 0.45 μm filter and dispensed into 1 mL aliquots. 250 μl was used to transduce HeLa cells, and 0.5μg/mL of Puromycin was used to select for integrated clones for 7 days.

## Trypan blue staining

25,000 cells were seeded in a 12-well plate and grown in Gibco DMEM supplemented with 10%FBS, 1X PenStrep, 2mM glutamine, and 1X Gibco NEAA and grown at 37°C with 5% $CO_2$. Cells were harvested at 24 hour timepoints for 72 hours. Cells were harvested by washing with 1X PBS, detached from the well with trypsin-EDTA (0.25%) for 3 min at 37°C then quenched with fresh complete media. Harvested cells were pelleted at 1000RPM for 5 min at room temperature. The supernatant was removed and the cell pellet was resuspended in 1X PBS, then pelleted again at 1000RPM for 5 min at room temperature. The supernatant was removed and the cell pellet was resuspended in 1 mL of PBS. 50μL of the 1X PBS cell suspension was mixed with 50μL of filter-sterilized Gibco 0.4% Trypan Blue Solution before counting on hemocytometer.

## Time-lapse microscopy

Images were collected every 20 min for 72 hours using Hamamatsu camera attached to the Olympus IX81 microscope at 10x magnification. Cells were maintained at 37°C and 5% $CO_2$. The exposure time was 250 ms for Brightfield and 100ms for TexasRed using Chroma filter ET560/40x (excitation) and ET630/75m (emission).

## Supporting information

**S1 Fig. Residual plot comparison of synchronous cells to asynchronous cells.**
(TIF)

**S2 Fig. Poisson and normal distribution EIV modeling comparison.**
(TIF)

**S3 Fig. Effect of population mean periodicity on cell cycle duration.**
(TIF)

**S4 Fig. Effect of population variance of cell cycle duration.**
(TIF)

**S5 Fig. Graphical hypothesis of LPS exposure to synchronous cell populations.**
(TIF)

**S6 Fig. Doubling time, cell density and viability assessment.**
(TIF)

**S7 Fig. Overall scheme of single-cell tracking of cell cycle duration.**
(TIF)

**S8 Fig. Normalized and curve fitted desynchronization rates from simulated model using experimental values.**
(TIF)

**S9 Fig. Normalized and curved fitted desynchronization rates for synchronized cells.**
(TIF)

**S1 Video. Fluorescent Microscopy of H2B-FT HeLa cells for LPS 0 μg/mL.**
(AVI)

**S2 Video. Fluorescent Microscopy of H2B-FT HeLa cells for LPS 0 μg/mL.**
(AVI)

**S3 Video. Fluorescent Microscopy of H2B-FT HeLa cells for LPS 1 μg/mL.**
(AVI)

**S4 Video. Fluorescent Microscopy of H2B-FT HeLa cells for LPS 1 μg/mL.**
(AVI)

**S1 Data. Coordinates for tracked cells in S1–S4 Videos.**
(XLSX)

## Acknowledgments

We thank Khai Nguyen and Bleris lab members for support and discussions.

## Author Contributions

**Conceptualization:** Chance M. Nowak, Leonidas Bleris.

**Data curation:** Chance M. Nowak, Tyler Quarton.

**Formal analysis:** Chance M. Nowak, Tyler Quarton, Leonidas Bleris.

**Funding acquisition:** Leonidas Bleris.

**Methodology:** Tyler Quarton, Leonidas Bleris.

**Project administration:** Leonidas Bleris.

**Supervision:** Leonidas Bleris.

**Validation:** Chance M. Nowak, Leonidas Bleris.

**Visualization:** Tyler Quarton.

**Writing – original draft:** Chance M. Nowak.

**Writing – review & editing:** Chance M. Nowak, Leonidas Bleris.

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
