## [Decision Letter · Decision Letter 0]

12 Jan 2023

Dear Dr. Bleris,

Thank you very much for submitting your manuscript "Impact of Variability in Cell Cycle Periodicity on Cell Population Dynamics" for consideration at PLOS Computational Biology.

As with all papers reviewed by the journal, your manuscript was reviewed by members of the editorial board and by several independent reviewers. In light of the reviews (below this email), we would like to invite the resubmission of a significantly-revised version that takes into account the reviewers' comments.

Referees 1 and 2 question what new biological insight the paper gives, which is more than the new experimental methodology, what is not in the focus of this journal. Please update the manuscript and show what new insight the new methodology provides.

We cannot make any decision about publication until we have seen the revised manuscript and your response to the reviewers' comments. Your revised manuscript is also likely to be sent to reviewers for further evaluation.

Sincerely,

Attila Csikász-Nagy

Academic Editor

PLOS Computational Biology

Kiran Patil

Section Editor

PLOS Computational Biology

Reviewer's Responses to Questions

**Comments to the Authors:**

Reviewer #1: The paper by Nowak et al. analyzes the effects of variability in the interdivision time on synchrony loss after a cell-cycle arrest, and uses modeling to propose a causal link. The data are of high quality and the conceptual approach is mostly correct. In particular, the application of the Kuiper´s test is interesting and provides a robust statistic to the analysis of cell-cycle synchrony.

The authors initially assumed the duration of S phase to be approximately 1/3 of the interdivision time. However, what the equation reflects is the half-time for DNA replication. As it is, the equation simulates a very short S phase, taking less than 30 min. I would suggest to increase S-phase length to 2-3 h as found by others. This could be attained by decreasing the e base to 1.1 (just a 10% of the interdivision time or so). It would be interesting to see what happens to synchrony loss rates when S-phase duration is increased to experimental levels and a realistic noise is added to the abovementioned base.

The paper finishes by showing the effects on synchrony loss when a paradigmatic signaling pathway is affected but, in my view, the direct impact of these data in the main conclusion of the paper is rather limited. Alternatively, fast loss of synchrony after cell division would allow daughter cells to face punctual stressors at different moments of the cell cycle and, hence, increase survival probability. I would suggest the authors to extend the discussion section considering these and other possible scenarios in which their findings would have a relevant functional role.

In summary, although it might have seemed obvious to the non-specialist researcher, the starting hypothesis had not been formally tested and this work will be interesting to those doing research in the interphase between cell proliferation and tissue organization and physiology.

Reviewer #2: General comments:

In this paper, the authors describe a combination of experiments and simulations to investigate the desynchronization properties in cervical cancer HeLa cells, starting from the G1/S boundary following double-thymidine block.

The authors' main conclusion is that cell cycle desynchronization rate is primarily sensitive to the variability of cell cycle duration.

While the authors cover an interesting topic, there are some major concerns that need to be addressed.

The link between cell cycle duration and (de)synchronization rate seems a straightforward connection, since having the same cell cycle duration is what defines the synchronous behavior within an homogeneous cells population following a block.

The authors experimentally show that an increase of cell cycle noise, increases cell cycle variability and desynchronizizes the population. The authors indicate that the factors or mechanisms that control cellular desynchronization remains largely unknown and their own phenomenological model is too simple to offer opportunities to explore any of the controlling factors. While the model is able to capture the experimental percentages of cells in various phases of the cell cycle, it does not seem to help in advancing the field in understanding the factors that control cellular desynchronization.

In the abstract the authors write that their results highlight an underexplored aspect in cell cycle research (i.e. using desynchronization rate of artificially synchronized in-phase cell populations as a proxy of the degree of variance in cell cycle periodicity), and they themselves leave that idea underexplored by not mentioning anything about this topic in any other part of the paper.

The Discussion section is extremely short, and mostly focused on background information about why cell cycle is a crucial process to study, while it fails to explain why the specific ideas from this paper make a significant contribution to the field.

The paper may contain experimental processes and methodologies suitable for publications in journals focused on the experimental protocols/methodologies, so if the authors believe that their experimental protocols are novel, we recommend to add more information about that aspect before resubmitting the paper to a different journal. Quoting the PLOS Computational Biology scope: research articles should demonstrate both methodological and scientific novelty, and provide profound new biological insights, and inclusion of experimental validation of a modest biological discovery through computation does not render a manuscript suitable for PLOS Computational Biology.

While the paper is well written and provides good general background and high-level context on cell cycle research, to publish this manuscript as novel research in PLOS Computational Biology, authors should add information about how the presented model or results advance cell cycle research in a profound way, what is the scientific novelty in this manuscript how their contribution helps advancing the scientific knowledge about the factors that lead to cell cycle variability.

Minor additional outstanding issues:

- duplicate citation (16, 30)

- please make sure NFkappaB/NFkB nomenclature is consistent

Reviewer #3: The goal of this manuscript is to investigate the desynchronization of human cells arrested in the same phase of their cell cycle. The authors propose the autosimilarity function (ASF), an elegant measure of cell cycle asynchrony, based on the cumulative distribution of cellular DNA amount. The measure is equivalent to the Kuiper two-sample test statistic. Next, a phenomenological mathematical model of DNA accumulation in cycling cells augmented with a stochastic term can produce ASF time-dependence as in the experiments. The model predicts that increasing the noise should accelerate desynchronization, which is verified experimentally by using lipopolysaccharide (LPS) to increase the noise of cell cycle periods.

Overall, this is an elegant, clearly presented, relatively simple yet interesting study that deserves publication. The manuscript should benefit from the authors addressing the following comments.

(1) Figure 1D: axis label and units are missing. An axis label would also be needed for panels 1b and 1c. Please ensure that all plots have axis labels and units.

(2) At least initially, the ASF might depend on the measurement intervals. For example, if accidentally the ASF was measured exactly at the time points where the red and green lines cross each other in Figure 1d, the ASF would be lower. It would be useful to include a rationale for choosing an optimal time interval for ASF measurements. This optimal interval probably depends on the cell cycle period, right?

(3) Do the data points in Figure 2c have error bars? It would be interesting to think of a statistical test for cells reaching asynchrony. This would require repeated measurements of ASF and testing if the ASF values of an initially synchronous population are significantly different from an asynchronous population’s ASF values.

(4) In Supplementary Figure 2, the effect of values selected from a normal versus Poisson distribution are compared. However, many Poisson distributions exist, depending on the distribution’s parameters. The parameters should be specified for both normal and Poisson distributions. Trying multiple parameters would be useful. For some parameter choices, the Poisson results should tend to be similar to the results obtained using normal-distributed values.

(5) Equation 5 and Figure 3: what aspects of the DNA synthesis does the noise term affect? Is it only the time of the uprise from dna0? Or also the slope of the rise?

(6) Is there a stronger justification for using the model in Equations 2 – 5 besides the shape of DNA accumulation over time? Are there other models of DNA accumulation versus cell cycle time in the literature? It would be useful to discuss this to understand the novelty of the approach in the context of other papers.

(7) “RelA, have shown to interact…”, probably a “have been” was intended here.

(8) While the LPS treatment increases the noise of cell cycle times, it seems to also affect their mean. Ideally, the average cell cycle time should stay unchanged. If not, then the change in the mean cell cycle time may affect the rate of desynchronization. This should be tested by modeling, doing a parameter scan for tau.

(9) It would be useful to develop a metric of how “fast” a cell population reaches asynchrony, and then apply it to Figure panels 4c, 4e. Yes, one curve is always below the other, but they start the same way, one below the other. So, it is like declaring a runner as the winner after giving him a head start. Would exponentials fit these curves? Could the exponent be a metric for the speed of approaching asynchrony?

(10) Regarding the independent effects of the noise and the mean, experimental approaches have been developed for their decoupled control (meaning that the noise changes while the mean does not). Prior work on this may be worth mentioning, see PMID:17189188 and PMID:31235692.

**Have the authors made all data and (if applicable) computational code underlying the findings in their manuscript fully available?**

Reviewer #1: Yes

Reviewer #2: Yes

Reviewer #3: Yes

PLOS authors have the option to publish the peer review history of their article (what does this mean?). If published, this will include your full peer review and any attached files.

Reviewer #1: No

Reviewer #2: No

Reviewer #3: No
---

## [Decision Letter · Decision Letter 1]

6 Apr 2023

Dear Dr. Bleris,

We are pleased to inform you that your manuscript 'Impact of Variability in Cell Cycle Periodicity on Cell Population Dynamics' has been provisionally accepted for publication in PLOS Computational Biology.

Best regards,

Attila Csikász-Nagy

Academic Editor

PLOS Computational Biology

Kiran Patil

Section Editor

PLOS Computational Biology

Reviewer's Responses to Questions

**Comments to the Authors:**

Reviewer #1: The authors have addressed my main comments and modified the paper accordingly

Reviewer #2: In the revised version of the paper (and in the point-to-point response) the authors demonstrate thoughtful care about the constructive feedback provided by all the reviewers.

The authors additional content included in the main manuscript provide good context on the presented ideas about the implications of cellular noise in cell synchrony and cell periodicity.

The authors addressed in a satisfactory way the points I previously raised.

While the simple phenomenological model has clear limitations, the study is overall well thought-out and presented so it could stimulate new research and advances in the field.

Reviewer #3: I would like to thank the authors for addressing the comments by all reviewers. This study will open new avenues of investigation into the sources and consequences of a new type of stochasticity. Therefore, I would like to recommend the publication of this revised version.

**Have the authors made all data and (if applicable) computational code underlying the findings in their manuscript fully available?**

Reviewer #1: None

Reviewer #2: Yes

Reviewer #3: Yes

PLOS authors have the option to publish the peer review history of their article (what does this mean?). If published, this will include your full peer review and any attached files.

Reviewer #1: No

Reviewer #2: No

Reviewer #3: No

---

## [Editor Report · Acceptance letter]

6 Jun 2023

PCOMPBIOL-D-22-01856R1 

Impact of Variability in Cell Cycle Periodicity on Cell Population Dynamics

Dear Dr Bleris,

I am pleased to inform you that your manuscript has been formally accepted for publication in PLOS Computational Biology. Your manuscript is now with our production department and you will be notified of the publication date in due course.

With kind regards,

Anita Estes
